# Sound Quality Performance of Orthogonal Antisymmetric Composite Laminates Embedded with SMA Wires

**DOI:** 10.3390/ma16093570

**Published:** 2023-05-06

**Authors:** Yizhe Huang, Jiangbo Hu, Jun Wang, Jinfeng Sun, Ying You, Qibai Huang, Enyong Xu

**Affiliations:** 1School of Mechanical Engineering, Hubei University of Technology, Wuhan 430068, China; yizhehuang@hbut.edu.cn (Y.H.); jiangbo_hu@hbut.edu.cn (J.H.); junwang@hbut.edu.cn (J.W.); 20051043@hbut.edu.cn (J.S.); youyong69@163.com (Y.Y.); 2State Key Laboratory of Digital Manufacturing Equipment and Technology, Huazhong University of Science and Technology, Wuhan 430074, China; qbhuang@hust.edu.cn; 3Dongfeng Liuzhou Motor Co., Ltd., Liuzhou 545005, China

**Keywords:** sound quality performance, orthogonal antisymmetric, shape memory alloys, composite laminates

## Abstract

Orthogonal antisymmetric composite laminates embedded with shape memory alloys (SMAs) wires have the potential to improve the sound quality of vibro-acoustics by taking advantage of the special superelasticity, temperature phase transition, and pre-strain characteristics of SMAs. In this research, space discretion and mode decoupling were employed to establish a vibro-acoustic sound quality model of SMA composite laminates. The association between the structural material parameters of SMA composite laminates and the sound quality index is then approached through methodologies. Numerical analysis was implemented to discuss the effects of SMA tensile pre-strain, SMA volume fraction, and the ratio of resin-to-graphite in the matrix on the vibro-acoustic sound quality of SMA composite laminates within a temperature environment. Subsequently, the sound quality test for SMA composite laminates is thus completed. The theoretically predicted value appears to agree well with the experimental outcomes, which validates the accuracy and applicability of the dynamic modeling theory and method for the sound quality of SMA composite laminates. The results indicate that attempting to alter the SMA tensile pre-strain, SMA volume fraction, and matrix material ratio can be used to modify loudness, sharpness, and roughness, which provides new ideas and a theoretical foundation for the design of composite laminates with decent sound quality.

## 1. Introduction

With the advancement of science and technology, the dynamic properties of mechanical systems, structures, and products’ have significantly improved, and their vibration and noise issues have gradually been effectively controlled. The focus of scholars on mechanical system vibration and noise has begun to change qualitatively, and product design has shifted from how to minimize noise to how to design for and control sound quality [1,2,3]. Sound quality is an acoustic indicator reflecting human subjective feelings, and includes loudness, roughness, sharpness, restlessness, etc. In actuality, attention to the sound quality of products in engineering has become very common. In the automotive field, pure electric vehicles, hydrogen fuel vehicles, and other new energy vehicles are the main development directions at present [4]. Although the engine is no longer the primary source of noise for new energy vehicles, the sound pressure level in such vehicles is significantly lower than that of conventional vehicles. However, the acoustical comfort of new energy vehicles has not been significantly improved, and the acoustical quality problem is more outstanding [5,6,7]. The main cause is the alteration of sound quality parameters such as loudness, sharpness, and roughness brought on by structural resonance and abnormal noise. Similar sound quality problems also exist in high-speed rail trains, civil aircraft, and household electrical appliances [8,9]. Therefore, the design and control of sound quality should not only address the issue of vibration and sound radiation intensity of machine equipment, but also how to effectively design and control its radiated sound quality.

In order to improve the radiated sound quality of products or structures, many scholars have carried out sound quality design and control research from the perspectives of structural dynamics and dynamic design, vibration and noise control, mode analysis, and active control [10,11]. Because sound quality indexes, such as loudness and sharpness, involve the composition of sound level and frequency, in sound quality design, it is more difficult than conventional noise control to consider how to control the sound level and coordinate the frequency composition of sound [12]. At present, the main methods for sound quality design and control include stiffness control, damping control, sound absorption and isolation, vibration, and vibration isolation [13]. By changing the geometrical dimensions of the structure and adjusting the structure mode, the sound quality design of the mode frequency can be realized. The sound radiation intensity of the structure can be effectively reduced, and the sound radiation quality can be improved by increasing the structure damping or adopting sound absorption and insulation measures. Sound quality is first applied in automotive products, including the whole vehicle, engine, exhaust system, transmission system, tires and doors, etc. [14,15]. For example, mode analysis is used to improve the sound quality of door closing by reducing the modal energy of the door, and comprehensive optimization technology of sound absorption and insulation is used to improve the sound quality of the interior of the vehicle [12,16]. Secondly, sound quality is more and more widely used in the household electrical industry, such as the sound quality design and optimization of air conditioners, fans, compressors, etc. [17,18]. In addition, some scholars have also studied the sound quality of aircraft, trains, etc., making important progress [6,19,20]. These results provide a basis for improving product design and sound quality characteristics from the perspective of sound quality. The composite laminated plate structure, as the most commonly used mechanical structure component in engineering, has been widely used in the fields of aerospace, ships, and transportation. Vibration and sound radiation of the composite laminate structure is one of the main noise sources of mechanical products and systems. Especially for large area composite laminate structure, the sound radiation intensity is high and there is multimode resonance, which makes the sound quality problem extremely complex.

As an intelligent material, shape memory alloys, which include NiTi, NiTiCu, and NiTiAl, possess the properties of superelasticity, temperature phase transition, shape memory effect, and pre-strain. They demonstrate promising application potential for noise and vibration control [21,22,23,24]. A new type of orthogonal antisymmetric composite laminate is formed by inserting SMAs into the matrix material of the laminate [25]. On the one hand, the elastic modulus of composites can be effectively adjusted to change the mechanical properties of composites [26,27,28,29]. On the other hand, the combination of the SMA material composition design and SMA tensile pre-strain design can further broaden the design range of the material structural modulus [30,31,32]. Therefore, the design adjustment of the mode frequency of SMA composite laminates can be realized, avoiding structural resonance, reducing vibration and sound radiation of composite plates, and improving the sound quality of composite laminates [33]. Firstly, a sound quality model of the vibration and sound radiation of SMA composite laminates is established by means of a space discretion and mode decoupling method. The relationship between the sound quality index and structural material parameters of SMA composite laminates is established theoretically. The effects of SMA tensile pre-strain, SMA volume fraction, and the composition ratio of graphite and resin in the matrix on the sound quality of vibration and sound radiation of SMA composite laminates under temperature were investigated via numerical analysis. Subsequently, a sound quality test for the SMA composite laminates was carried out. Compared with the theoretical calculation, the theoretical predicted value fits well with the experimental results, which verifies the correctness and validity of the dynamic modeling theory and method for the sound quality of SMA composite laminates. It provides new ideas and theoretical basis for sound quality design of composite panels and has important scientific value and research significance.

## 2. Formulation of Sound Quality Evaluation

The differential equation of vibration of a composite laminate embedded with SMA wires under external excitation can be expressed as
(1)$${\left[L\right]}_{\ell \times \ell }{\left\{\chi (x,y,t)\right\}}_{\ell \times 1}={\left\{F(x,y,t)\right\}}_{\ell \times 1}$$
where ${\left\{F(x,y,t)\right\}}_{\ell \times 1}$ refers to the excitation force vector and ℓ represents the vector dimension. Kirchhoff’s theory is used for a composite laminates model where ℓ is taken as 3. ${\left\{\chi (x,y,t)\right\}}_{\ell \times 1}$ denotes the displacement response vector. ${\left[L\right]}_{\ell \times \ell }$ is an operator which contains the stiffness operator ${\left[K\right]}_{\ell \times \ell }$, mass operator ${\left[M\right]}_{\ell \times \ell }$, and damping operator ${\left[C\right]}_{\ell \times \ell }$, i.e.,
(2)$${\left[L\right]}_{\ell \times \ell }{={\left[K\right]}_{\ell \times \ell }+{\left[C\right]}_{\ell \times \ell }+\left[M\right]}_{\ell \times \ell }$$

The operator element can be obtained from the operator matrix using the dynamic equation of orthogonal antisymmetric composite laminates embedded with SMA wires [34].
(3)$$\left[\begin{array}{ccc} L_{11} & L_{12} & L_{13}\\ L_{21} & L_{22} & L_{23}\\ L_{31} & L_{32} & L_{33} \end{array}\right]=\left[\begin{array}{ccc} A_{11}\frac{\partial^{2}}{\partial x^{2}}+A_{66}\frac{\partial^{2}}{\partial y^{2}} & \left(A_{12}+A_{16}\right)\frac{\partial^{2}}{\partial x\partial y} & -B_{11}\frac{\partial^{3}}{\partial x^{3}}\\ \left(A_{12}+A_{16}\right)\frac{\partial^{2}}{\partial x\partial y} & A_{66}\frac{\partial^{2}}{\partial x^{2}}+A_{11}\frac{\partial^{2}}{\partial y^{2}} & B_{11}\frac{\partial^{3}}{\partial y^{3}}\\ -B_{11}\frac{\partial^{3}}{\partial x^{3}} & B_{11}\frac{\partial^{3}}{\partial y^{3}} & K_{33}+C+M \end{array}\right]$$
where
(4)$$\begin{array}{c} L_{33}{=D}_{11}\frac{\partial^{4}}{\partial x^{4}}+D_{22}\frac{\partial^{4}}{\partial y^{4}}+2\left(D_{12}+2D_{66}\right)\frac{\partial^{4}}{\partial x^{2}\partial y^{2}}-\left(N_{SMAC,x}^{rec}-N_{m,x}^{T}\right)\frac{\partial^{2}}{{\partial x}^{2}}\\ -\left(N_{SMAC,y}^{rec}-N_{m,y}^{T}\right)\frac{\partial^{2}}{{\partial y}^{2}}+\lambda \frac{\partial }{\partial t}+I\frac{\partial^{2}}{\partial t^{2}} \end{array}$$

The displacement variables are spatially discretized by means of displacement functions so that the generalized displacement variable corresponding to Kirchhoff’s theory under simply supported boundary conditions can be set as
(5)$$\left\{\chi (x,y,t)\right\}=\left\{\begin{array}{c} \sum_{m=1}^{\infty }\sum_{n=0}^{\infty }U_{mn}sin\frac{m\pi x}{a}cos\frac{n\pi y}{b}\\ \sum_{m=0}^{\infty }\sum_{n=1}^{\infty }V_{mn}cos\frac{m\pi x}{a}sin\frac{n\pi y}{b}\\ \sum_{m=1}^{\infty }\sum_{n=1}^{\infty }W_{mn}sin\frac{m\pi x}{a}sin\frac{n\pi y}{b} \end{array}\right\}e^{i\omega t}$$

It is assumed that the composite laminate is subjected to a simple harmonic external load in the direction perpendicular to the X–Y plane, which can be expressed as
(6)$$\left\{F_{mn}\right\}={\left\{0,0,F_{mn}^{*}\right\}}^{T}$$

The amplitude of the (*m*,*n*)-order modal vibration velocity of composite laminates can be formulated as [34]
(7)$${\hat{U}}_{mn}\left(x,y\right)=j\omega F_{mn}^{*}/\left\{\bar{\rho }\left[\left(\omega_{mn}^{2}-\omega^{2}\right)+j\omega \lambda /\omega_{mn}\sum_{k=1}^{N}{\bar{\rho }}^{\left(k\right)}\right]\right\}sin\frac{m\pi x}{L_{a}}sin\frac{n\pi y}{L_{b}}$$
where λ denotes the equivalent viscous damping factor and ${\bar{\rho }}^{\left(k\right)}$ represents the area density of the *k*th layer. The $\omega_{mn}$ is the natural frequency of SMA composite laminates, which can be solved using an eigenvalue equation.

For orthogonal antisymmetric composite laminates embedded with SMA wires, any unit in composite laminates can be deemed as a source of external vibratory acoustic radiation. The radiation sound pressure of the composite laminates is obtained by using Rayleigh’s integral for the radiating sound pressure of all units. Then, the radiated sound intensity of the composite laminate can be expressed as
(8)$$E_{mn}\left(x,y\right)=\frac{1}{2}R_{e}\left[j\frac{k\rho_{0}c_{0}}{2\pi }\iint_{S}\frac{e^{-jkr}}{r}{\hat{U}}_{mn}\left(x^{\prime },y^{\prime }\right)ds^{\prime }{\hat{U}}_{mn}^{*}\left(x,y\right)\right]$$
where k, $\rho_{0}$, and $c_{0}$ are the number of sound waves, air density, and sound velocity, respectively. r is the distance from the sound radiation unit $ds^{\prime }$ to the vibration unit ds. The ${\hat{U}}_{mn}^{*}\left(x,y\right)$ is the conjugate of ${\hat{U}}_{mn}\left(x^{\prime },y^{\prime }\right)$.

The objective evaluation indexes of sound quality mainly include loudness, sharpness, modulation, tonality, and articulation, among which loudness, sharpness and roughness are the most important ones. Loudness is a psychoacoustic index of the overall loudness of the sound to the human ear. For the loudness on a single critical bandwidth, that is, the ratio loudness, the deduced SMA composite laminates’ acoustic radiation ratio loudness can be expressed as
(9)$$F_{N^{\prime }}\left(z,X\right)=0.08{\left(\frac{E_{TQ}}{E_{0}}\right)}^{0.23}\left[{\left(0.5+0.5\frac{E(z,X)}{E_{TQ}}\right)}^{0.23}-1\right]$$
where $X=\left(x_{1},x_{2}\cdots x_{k}\right)$ is the design parameter vector of the SMA composite laminates and $E_{0}$ is the stimulus intensity corresponding to the physical sound intensity reference value $10^{-12}\;W/m^{2}$. $E_{TQ}\left(f\right)$ is the stimulus intensity produced by the static threshold with a value of $10^{0.364{\left(f/1000\right)}^{-0.8}-12}$.

The total loudness is the integral of the ratio loudness in the whole characteristic frequency band, which can be expressed as Equation (10).
(10)$$F_{N}\left(X\right)=\int_{0}^{24Bark}F_{N^{\prime }}\left(z,X\right)dz$$

Sharpness is a psychological index to measure the amount of high-frequency components in the sound, which reflects the distribution of high-frequency components in the spectrum of sound radiation. The calculation methods of sharpness in this study mainly include the Aures method and von Bismark method. The calculation for the Aures method is expressed as
(11)$$F_{S}^{Aure}\left(X\right)=0.0088\frac{\int_{0}^{24Bark}F_{N^{\prime }}\left(z,X\right)e^{0.171z}dz}{log\left(\frac{F_{N}\left(X\right)}{20}+1\right)}$$

The von Bismark calculation is expressed as
(12)$$F_{S}^{Bismarke}\left(X\right)=\left\{\begin{array}{c} 0.11\frac{\int_{0}^{24Bark}F_{N^{\prime }}\left(z,X\right)zdz}{F_{N}\left(X\right)},z<14\\ 0.11\frac{\int_{0}^{24Bark}F_{N^{\prime }}\left(z,X\right)z\left[1+\frac{3}{1000}\left(z-14\right)\right]dz}{F_{N}\left(X\right)},z\geq 14 \end{array}\right.$$

The two sharpness models mentioned above reflect the distribution of high-frequency components in composite laminates in regard to the total noise from different aspects. Therefore, the result calculated using the Bismark method is more faithful to the reality of human feelings.

The roughness model of SMA composite laminates can be expressed as
(13)$$F_{R}\left(X\right)=0.3\frac{f_{mod}}{1000}\int_{0}^{24Bark}20lg\frac{{F_{N^{\prime }}\left(z,X\right)}_{max}}{{F_{N^{\prime }}\left(z,X\right)}_{min}}dz$$

For the vibration and radiation noise of the composite laminates, the time-domain masking depth is difficult to analytically express as a function of the characteristic frequency band scale. In the calculation, the change in the characteristic loudness of each critical frequency band at 1 m from the radiation surface of the composite laminate is used as a substitute.

## 3. Results and Discussion

The research objects of this section are 10-layer SMA rectangular orthogonal composite laminates simply supported on four sides, whose length, width, and thickness are 0.4 m, 0.3 m, and 0.008 m, respectively. Young’s modulus of elasticity and pre-strain recovery stress of SMAs are derived from [35]. Poisson ratio, density, and thermal expansion coefficient of the shape memory alloys are 0.3, 6450 kg/m^2^, and 10.26 × 10^−6^/°C, respectively. Shape memory alloys are embedded in each layer at a 20% volume fraction with a matrix material ratio of 1:9 of graphite and diphenol propane epoxy resins. Sound quality was analyzed in terms of temperature, SMA pre-strain, SMA volume fraction, and matrix material ratio. The vibro- acoustic radiation sound quality of orthogonal antisymmetric composite laminates embedded with SMA wires is schematically depicted in Figure 1. In response to an external excitation force, the composite laminates produce vibro-acoustic radiation noise. At various temperatures, the SMAs in the laminates are in various phases, and this affects the vibro-acoustic radiation sound quality.

Two representative temperatures, 25 °C at room temperature and 100 °C at high temperature, were selected as the temperature environment conditions. The SMA wires at 25 °C and 100 °C were in the complete martensite phase and the complete austenite phase, respectively. Under the condition that the volume ratio of graphite-to-epoxy resin is 1:9 and the volume fraction of the SMAs is 20%, the sound quality of the SMA composite plate is calculated under three conditions: tensile pre-strain of SMA wire at 1%, 3%, and 5%, respectively. Figure 2a–d are Bark diagrams of the loudness and roughness of the corresponding radiated sound. Table 1 shows the loudness, sharpness, and roughness of the radiated sound over a wide band.

From the comprehensive comparison of Figure 2 and Table 1, the variation in the pre-strain has relatively little effect on sound loudness and roughness at a room temperature of 25 °C, which is basically consistent with the sound radiation law. Loudness, sharpness, and roughness decrease slightly when SMA pre-strain reaches 5%. SMA pre-strain has an obvious effect on sound quality at 100 °C. The loudness, sharpness, and roughness increase with the increase in pre-strain. The main reason is that with the increase in SMA pre-strain, the stiffness and mode frequency of the SMA composite plate caused by pre-strain recovery stress increase, resulting in the increase in the high-frequency component. That is, pre-strain has a greater impact on loudness and roughness.

For the influence of the SMA volume fraction on the sound quality of the composite plates, four kinds of SMA volume fractions (10%, 20%, 30%, and 40%) were considered, and sound quality analysis was carried out under two temperatures: 25 °C at room temperature and 100 °C at high temperature. The Bark diagram of sound quality is shown in Figure 3a–d, and Table 2 shows the results of the total sound quality index.

From Figure 3a–d and Table 2, the loudness and sharpness decrease with the increasing volume fraction of the SMAs at both temperatures. Loudness reduction can be attributed to two reasons: one is the decrease in sound radiation power, and the other is the increase in low-frequency components. The decrease in sharpness with volume fraction further indicates that the increase in low-frequency components in sound radiation has an effect on the decrease in loudness. The roughness decreases gradually with increasing volume fraction at 25 °C. The variation trend of roughness at 100 °C is opposite to that at 25 °C. The total roughness is smaller than that at 25 °C. This indicates that the sound radiation roughness of the SMA composite plate will be improved with increasing temperature.

Considering the large modulus of graphite in the matrix, the proportion of graphite content has a significant influence on the overall stiffness of the SMA composite plate. When the volume fraction of SMA fiber is 20%, the volume fraction of matrix is 80%. The volume fraction ratio *V_g_*:*V_e_* of graphite and epoxy resin in the matrix was 1:9, 2:8, 3:7, and 4:6, respectively. Figure 4a–d show the effects of the volume fraction ratios of different matrix materials on the sound radiation loudness and roughness of SMA composite plates at 25 °C and 100 °C, respectively. Table 3 shows the results of the total sound power and sound quality.

From Figure 4a–b and Table 3, the modulus of the SMAs is relatively less at 25 °C, and the modulus of the matrix material plays a leading role. With the increase in graphite proportion, the peak of loudness also moves towards a high Bark reading. In the frequency range of 0 Hz–1000 Hz, the Bark numbers with loudness values greater than 9 are 4, 3, 2, and 2, respectively when the substrate material ratio $V_{g}/V_{e}$ is 1:9, 2:8, 3:7, and 4:6. Similarly, the Bark numbers with loudness values greater than 9 are 2, 3, 1 and 1, respectively, at 100 °C. In conjunction with Table 3, the greater the number of Barks with high loudness values, the greater the broadband loudness values. The sharpness depends more on the specific gravity of the high loudness value in the high-frequency part of the characteristic band spectrum. The response values for 6 Bark, 7 Bark, and 9 Bark are 10.338, 10.139, and 10.571, respectively, when the ratio of $V_{g}/V_{e}$ is 2:8 at 25 °C. The high loudness value distributes more in high-frequency components and the corresponding sharp value is higher. Similarly, the sharpness value of the matrix material at 100 °C is higher than that at $V_{g}/V_{e}$ = 2:8.

In consideration of Figure 4c,d and Table 3, the sound radiation roughness of different matrix material volume fractions at 25 °C is higher than that at 100 °C. The corresponding sound radiation roughness of the base material at 25 °C is 1:9 for $V_{g}/V_{e}$, which is significantly higher than that of other conditions. The main reason is that the roughness value of the characteristic band of 2 Bark is 0.445 Asper, which is significantly higher than that of other characteristic bands, with the roughness value of other characteristic bands being less than 0.300 Asper. Therefore, the range between the maximum and minimum of characteristic loudness in the characteristic band of 2 Bark is larger than that in other characteristic bands, resulting in a large time-domain masking depth. From Equation (13), if the modulated audio rate is determined to be 70 Hz, the masking depth is proportional to the sound radiation roughness.

## 4. Experimental Verification

The sound power is measured using the sound intensity scanning method. The vibration exciter is primarily used for point force excitation in the sound quality experiment of the SMA composite laminates. The measurement system includes the B&K3560 Acquisition System, B&K 4197 Sound Intensity Probe, B&K 4825 Exciter, and Pulse 3560 Signal Test and Analysis System as shown in Figure 5. On one side of the SMA composite laminates, the vibration exciter is positioned, and the sound intensity probe is employed to measure sound intensity on the opposite side of the laminates. The vibration exciter is packaged in a closed sound insulation anechoic box with sound-absorbing materials on the inner wall during the experiment to avoid the interference of the sound diffraction of the vibration exciter on the sound intensity probe measurement.

Step 1: The center of the laminates serves as the vibration exciter’s excitation point. The frequency measurement range is within 1000 Hz, and the sine sweep signal is used as the experimental excitation signal. The exciting force at each measuring frequency is maintained constant by adjusting the output voltage and the power amplifier’s gain value.

Step 2: In accordance with the laminate’s size, determine the grid points for measuring sound intensity. Mark the SMA composite laminates with a 9 × 9 grid in a manner similar to the modal test. Sixty-three measuring points on the grid are scanned via the sound intensity probe, with a 5-s measurement time for each point.

Step 3: The radiated sound intensity of the composite laminates is obtained in Pulse3560 Signal Test and Analysis System and converted into the radiated sound power.

Step 4: Repeat steps 1 through 3 to obtain the measured radiated sound power of the composite laminate.

The ● line in Figure 6 represents the sound radiation power of the composite laminates as measured using the sound intensity method. The theoretical results apply the theory derived in this paper to calculate the sound radiation power of the SMA composite laminates, as shown by the □ line.

Figure 6 depicts a general consistency between the theoretical and experimental results for the sound radiation power level of SMA composite laminates. The peak sound radiation power level of SMA composite laminates has a theoretical prediction value that is slightly higher than the experimental result. The connection damping between the layers in the laminates is not taken into account theoretically; however, there is some connection damping between the layers in the experimental test sample, which decreases the sound radiation intensity of the SMA composite laminates experimental test.

Additionally, a modal error is primarily responsible for the variation in the peak frequency of acoustic radiation. The experimental values for the peak frequency of the sound radiation of the third-, fourth-, and fifth-orders are lower than the theoretical values. That is to say, the actual stiffness of the composite laminates is smaller than the theoretical value, which may be due to the fact that the bonding strength of the SMA wires when applied to the matrix during the fabrication of the composite laminates does not reach the desired level, resulting in the slightly lower actual stiffness of the SMA composite laminates [33].

The sound power is converted into sound loudness level, sharpness, and roughness, the theoretical prediction values and experimental results are shown in Table 4.

As per Table 4, the loudness level error of the SMA composite laminates between the experimental results and the theoretical prediction is less than 5%. Since the sharpness is mainly related to the distribution of sound radiation frequency, and the theoretical prediction value is further converted based on the theoretical prediction value of loudness, this increases the theoretical prediction error of sharpness and roughness to a certain extent. The error between the theoretical prediction of the sharpness and roughness of the SMA composite laminates and the experimental results is within 10%. Therefore, the experimental results demonstrated the accuracy and efficiency of the theoretical approaches covered in this paper.

## 5. Conclusions

This study establishes a model for the sound quality of orthogonal antisymmetric composite laminates embedded with SMA wires and conducts a thorough analysis of the SMA composite laminates’ sound quality performance. The following are the main conclusions: (1)At 25 °C ambient temperatures, a variation in SMA tensile pre-strain has little effect on sound loudness, sharpness, or roughness. The loudness and roughness are slightly reduced, while the SMA tensile pre-strain is increased to 5%. At 100 °C, loudness, sharpness, and roughness are all positively correlated with the tensile pre-strain of the SMAs.(2)Loudness and sharpness at temperatures of 25 °C and high temperatures of 100 °C significantly decline as the SMA volume fraction increases. At 25 °C, the roughness gradually decreases as the volume fraction increases. At 100 °C, the roughness increases gradually with the volume fraction, though less so than at 25 °C.(3)In the frequency range of 1000 Hz, the number of Barks with loudness greater than 9 for different matrix materials at 25 °C is higher than that at the high temperature of 100 °C. When the ratio of the matrix material is 2:8, the high loudness value distributes more in the high-frequency component and the sharp value is higher. The sound radiation roughness of different matrix material volume fractions at 25 °C is higher than that at 100 °C.

In conclusion, the high temperature environment is more advantageous for SMA wires to improve sound quality for composite laminates that embed SMA wires. Through lessening the SMA tensile pre-strain and increasing the volume fraction of SMAs, there is further potential to significantly improve the loudness and roughness. By constructing the matrix material ratio properly, the distribution of high-frequency loudness components can be altered, and sharpness can be improved.

## Figures and Tables

**Figure 1 materials-16-03570-f001:**
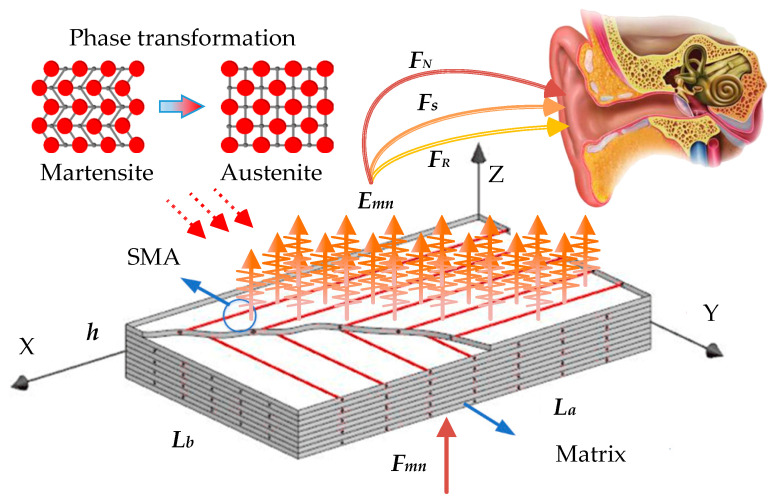
Vibration and sound radiation quality schematic diagram of orthogonal antisymmetric composite laminates embedded with SMA wires.

**Figure 2 materials-16-03570-f002:**
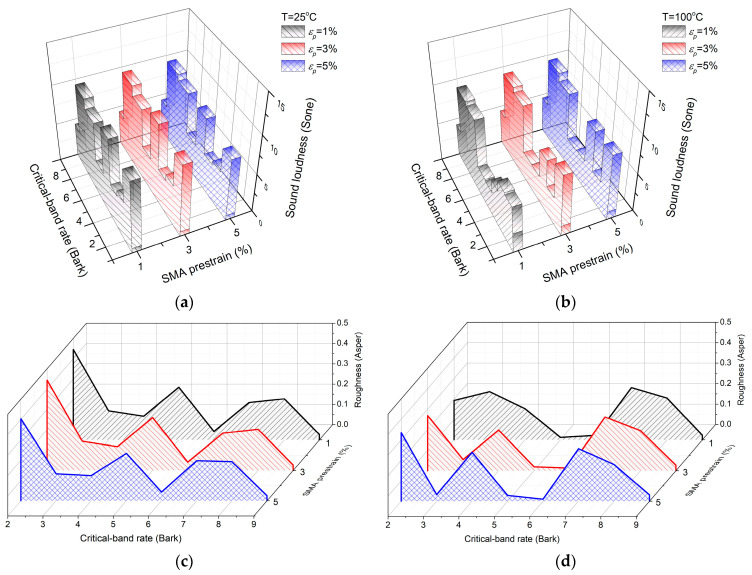
Sound radiation quality of SMA composite plates under different pre-strain conditions: (**a**) sound radiation loudness at 25 °C under different pre-strain conditions; (**b**) sound radiation loudness at 100 °C under different pre-strain conditions; (**c**) sound radiation roughness at 25 °C under different pre-strain conditions; (**d**) sound radiation roughness at 100 °C under different pre-strain conditions.

**Figure 3 materials-16-03570-f003:**
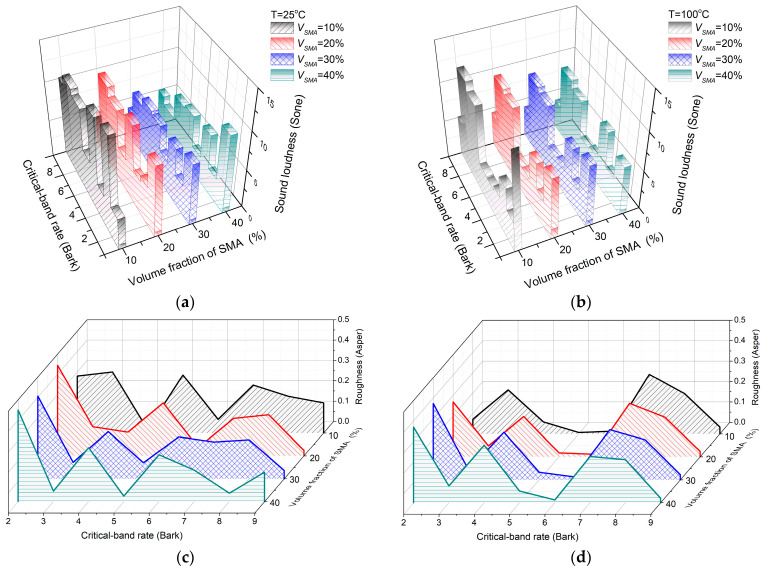
Sound radiation quality of SMA composite plates at different SMA volume fractions: (**a**) sound radiation loudness at 25 °C at different SMA volume fractions; (**b**) sound radiation loudness at 100 °C at different SMA volume fractions; (**c**) sound radiation roughness at 25 °C at different SMA volume fractions; (**d**) sound radiation roughness at 100 °C at different SMA volume fractions.

**Figure 4 materials-16-03570-f004:**
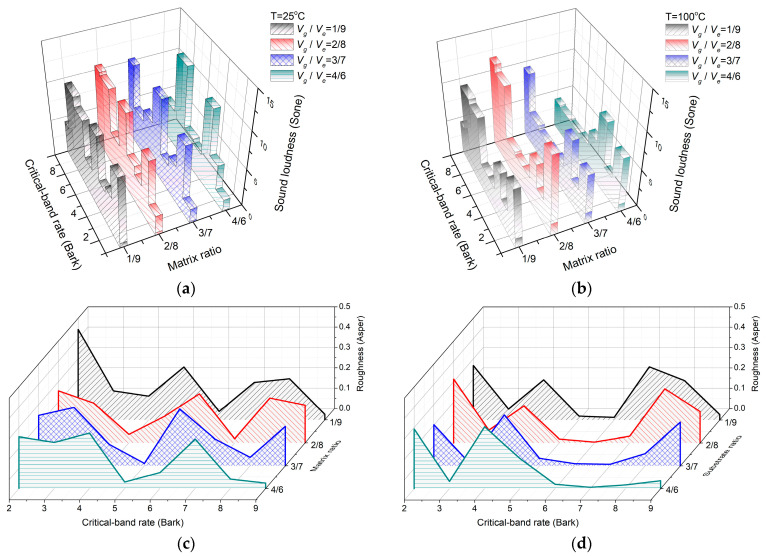
Sound radiation quality of SMA composite plates at different volume ratios of matrix material: (**a**) sound radiation loudness at 25 °C with different volume ratios of matrix material; (**b**) sound radiation loudness at 100 °C at different volume ratios of matrix material; (**c**) sound radiation roughness at 25 °C at different volume ratios of matrix material; (**d**) sound radiation roughness at 100 °C at different volume ratios of matrix material.

**Figure 5 materials-16-03570-f005:**
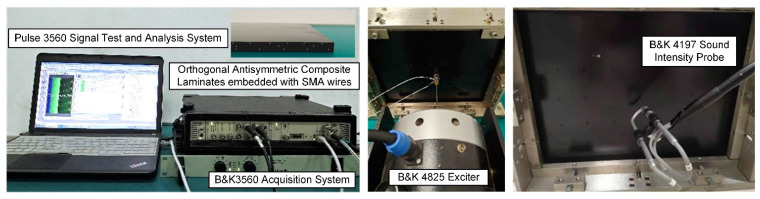
The system for measuring sound power of SMA composite laminates using sound intensity scanning.

**Figure 6 materials-16-03570-f006:**
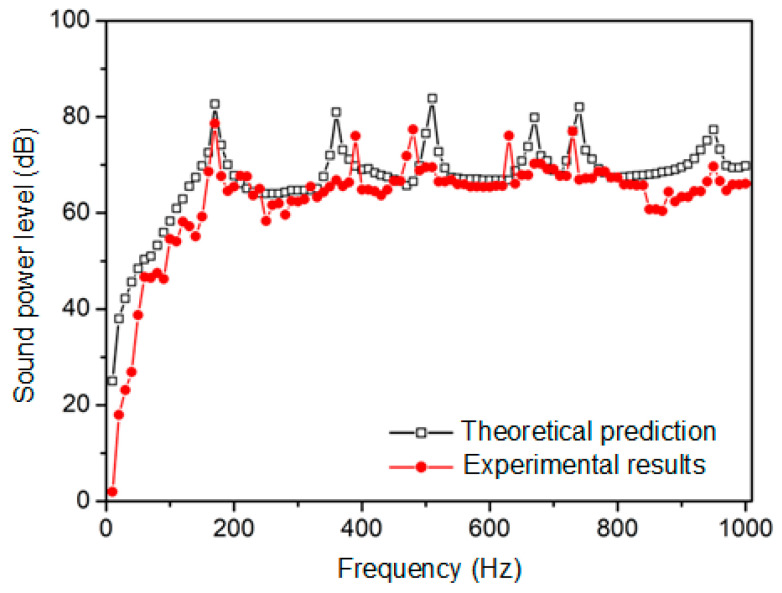
Theoretical prediction and experimental results of sound power level of SMA composite laminates.

**Table 1 materials-16-03570-t001:** Sound quality of SMA composite laminates subjected to different tensile pre-strain.

10 Hz–1 kHz		Tensile Pre-Strain at T = 25 °C			Tensile Pre-Strain at T = 100 °C		
(1 Bark–9 Bark)		$\varepsilon_{p}=$ 1%	$\varepsilon_{p}=$ 3%	$\varepsilon_{p}=$ 5%	$\varepsilon_{p}=$ 1%	$\varepsilon_{p}=$ 3%	$\varepsilon_{p}=$ 5%
Loudness		58.11	58.03	55.81	50.74	50.97	53.35
Sharpness	Aures	2.36	2.35	2.35	2.24	2.27	2.30
	von Bismark	0.59	0.59	0.60	0.59	0.60	0.60
Roughness		1.41	1.41	1.35	1.09	1.03	1.10

**Table 2 materials-16-03570-t002:** Sound quality of SMA composite laminates subjected to different volume fraction of SMA.

10 Hz–1 kHz		Volume Fraction of SMA at T = 25 °C				Volume Fraction of SMA at T = 100 °C			
(1 Bark–9 Bark)		$V_{s}=$ 10%	$V_{s}=$ 20%	$V_{s}=$ 30%	$V_{s}=$ 40%	$V_{s}=$ 10%	$V_{s}=$ 20%	$V_{s}=$ 30%	$V_{s}=$ 40%
Loudness		71.80	58.11	49.43	47.95	63.74	50.97	48.03	45.44
Sharpness	Aures	2.75	2.36	2.15	2.06	2.37	2.27	2.19	2.15
	von Bismark	0.63	0.59	0.58	0.56	0.55	0.61	0.60	0.61
Roughness		1.51	1.41	1.41	1.36	0.88	1.03	1.16	1.26

**Table 3 materials-16-03570-t003:** Sound quality of SMA composite laminates subjected to different volume ratios of matrix material.

10 Hz–1 kHz		Volume Ratios of Matrix Material at T = 25 °C				Volume Ratios of Matrix Material at T = 100 °C			
(1 Bark–9 Bark)		$V_{g}/V_{e}=$ 1/9	$V_{g}/V_{e}=$ 2/8	$V_{g}/V_{e}=$ 3/7	$V_{g}/V_{e}=$ 4/6	$V_{g}/V_{e}=$ 1/9	$V_{g}/V_{e}=$ 2/8	$V_{g}/V_{e}=$ 3/7	$V_{g}/V_{e}=$ 4/6
Loudness		58.11	55.77	51.53	46.57	50.97	55.17	43.65	41.13
Sharpness	Aures	2.36	2.58	2.43	2.20	2.27	2.46	2.13	1.85
	von Bismark	0.59	0.67	0.65	0.62	0.60	0.62	0.59	0.53
Roughness		1.41	1.28	1.29	1.17	1.03	1.03	0.79	0.86

**Table 4 materials-16-03570-t004:** Theoretical prediction and experimental results of sound quality of SMA composite laminates.

Sound Quality	Theoretical Prediction	Experimental Results	Error
Loudness level (Phon)	95.76	92.32	3.73%
Sharpness (Acum)	2.08	1.96	6.12%
Roughness (Asper)	0.98	0.93	5.38%

## Data Availability

Data available on request.

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
