# Peer review of "Sound Quality Performance of Orthogonal Antisymmetric Composite Laminates Embedded with SMA Wires"

_materials, 2023, doi:10.3390/ma16093570_

Round 1

Reviewer 1 Report

Sound Quality Performance of Orthogonal Antisymmetric 2 Composite Laminates embedded with SMA wires

1. The authors need to explain the novelty of the current work.

2. While using the term SMA for the first time in abstract, please expand it.

3. What is the basis for ‘Formulation of sound quality evaluation’?

4. Figure 2 and 3 (a, b, c, d) quality to be enhanced.

5. Please incorporate the picture of the actual sample fabricated, showing SMA wires.

6. The authors can refer and cite this work for improved understanding: https://doi.org/10.1016/j.jallcom.2022.166027, 10.1088/0964-1726/14/4/001

7. What is the use of having SMA embedded in sound quality? Did the authors do a test without SMA embedded?

Reviewer 2 Report

The paper applies very specific and sophisticated methods for acoustics.
The authors should better explain how they applied these methods and how the acoustic measurements are performed and how to interpret the data.
It is necessary to clarify the purpose of the paper (I think it is interesting and innovative) the aim is to understand the acoustic effects of the study material, not to apply the techniques of piscoacoustic.
The concepts expressed in the paper are for specialists in the acoustics sector, and not all acoustics researchers are aware of these concepts (I take the liberty of writing because I apply them in the acoustic quality of interior spaces).
This type of analysis is performed in the automotive sector to search for the best and most acceptable sound emitted by cars both internally and externally.
The analysis is performed in the laboratory with dedicated software (the formulas shown in the paper leave their time!).
There is no discussion section.
So I asked the authors to write the paper better and make it more readable also for a non-acoustic audience in order to share it with a greater number of people.

Reviewer 3 Report

This paper presents a study of orthogonal antisymmetric composite laminates, in which SMA wires are embedded. It is shown that they have the potential to improve vibroacoustic sound quality by taking advantage of the special superelasticity, temperature phase transition, and pre-strain characteristics of SMA. The article is written in good style. The research demonstrated that a high-temperature environment is more favorable for SMA wires to improve the sound quality of composite laminates in which SMA wires are embedded. By reducing SMA pre-strain during stretching and increasing the volume fraction of SMA, there is further potential for significant improvements in loudness and roughness. With proper matrix material ratio construction, the distribution of high frequency loudness components can be altered and sharpness can be improved. The materials used are shape memory materials that can make the phase transition from the martensite phase to the austenite phase. The effect of the matrix, that is, the ratio of graphite to epoxy resin in it, has also been demonstrated.

The article has implications for improving sound quality and noise reduction systems, which are relevant to the automotive, aircraft, and railroad industries. However, the content of the article does not fit well with the content of Materals journal. The content of the article is more in line with the content of Acoustics journal and more in the direction of industrial acoustics.

On the whole, however, a comment should be made on the article:

(1) In the text, be sure to give the deciphering of SMA.

(2) Most people live in temperature conditions from 0°C to 30°C, so it is advisable to decide the optimal solution in this temperature range.

(3) The type of epoxy used should be described.

In terms of design, too, a number of technical remarks should be noted:

(1) line 38: “present[4]” replaced by “present [4]”

(2) line 51: “control[10, 11]” replaced by “control[10, 11]”

(3) line 55: “sound[12]” replaced by “sound [12]”

(4) line 68: “etc[17, 18]” replaced by “etc [17, 18]”

(5) line 70: “progress[6, 19]” replaced by “progress [6, 19]”

(6) line 79: “characteristics[20-22]” replaced by “characteristics [20-22]”

(7) line 81: “laminate[23]” replaced by “laminate [23]”

(8) line 82: “composites[24-27]” replaced by “composites [24-27]”

(9) line 84: “modulus[28-30]” replaced by “modulus [28-30]”

(10) line 87: “laminates[31]” replaced by “laminates [31]”

Round 2

Reviewer 1 Report

Accept

Author Response

Thank you very much for your suggestion. We will be committed to further improving the paper.

Reviewer 2 Report

improve referencese

Author Response

Thank you very much for your suggestion. We will carefully review and improve the reference section.

Reviewer 3 Report

The authors have made corrections. But in the future it would be more appropriate to publish articles on this subject in acoustical journals.

Author Response

Thank you very much for your suggestion. In the future, we will submit the relevant subject papers to the journal of acoustics.